# On the Generalization of Gradient-based Neural Network Interpretations

## Abstract

Feature saliency maps are commonly used for interpreting neural network predictions. This approach to interpretability is often studied as a post-processing problem independent of training setups, where the gradients of trained models are used to explain their output predictions. However, in this work, we observe that gradient-based interpretation methods are highly sensitive to the training set: models trained on disjoint datasets without regularization produce inconsistent interpretations across test data. Our numerical observations pose the question of how many training samples are required for accurate gradient-based interpretations. To address this question, we study the generalization aspect of gradient-based explanation schemes and show that the proper generalization of interpretations from training samples to test data requires more training data than standard deep supervised learning problems. We prove generalization error bounds for widely-used gradient-based interpretations, suggesting that the sample complexity of interpretable deep learning is greater than that of standard deep learning. Our bounds also indicate that Gaussian smoothing in the widely-used SmoothGrad method plays the role of a regularization mechanism for reducing the generalization gap. We evaluate our findings on various neural net architectures and datasets, to shed light on how training data affect the generalization of interpretation methods.

## 1 Introduction

Multi-layer neural network (NN) models have achieved revolutionary success in computer vision problems including image recognition (Krizhevsky et al., 2017), object detection (Zhao et al., 2019), and medical image processing (Litjens et al., 2017). This success is primarily due to the enormous capacity of NNs as well as their impressive generalization performance from training samples to unseen data. In other words, not only do massive NNs perform almost perfectly in predicting the label of training samples, but also they maintain their satisfactory training performance on test data unobserved during the NN model's training. The mysterious generalization success of deep learning models has attracted a lot of attention in the machine learning community.

While NNs achieve great prediction performance over standard computer vision datasets, their deployment in real-world applications such as self-driving cars and machine-based medical diagnostics requires a reliable interpretation of their predictions. Such interpretation of these large-scale models will help domain experts understand the basis of their predictions to further improve and robustify the prediction model. Over the recent years, several algorithms have been developed to give such an interpretation, including the widely-used gradient-based feature saliency maps such as the simple gradient (Baehrens et al., 2010; Simonyan et al., 2013), integrated gradients (Sundararajan et al., 2017), and SmoothGrad (Smilkov et al., 2017) methods. These gradient-based algorithms are based on the first-order derivative of the NN model's score function with respect to the input variables, which reveal the features with a major impact on the model's prediction.

While the gradient-based interpretation methods have found many applications in computer vision problems, the theoretical understanding of the underlying factors contributing to their performance is still largely inadequate. Specifically, the generalization aspect of standard interpretation methods has not been studied in the literature, and it remains unclear how many training samples are required to ensure a bounded variance of gradient-based explanation maps with respect to a random training set and stochasticity of the training process. Therefore, characterizing the sample complexity of es-

timating saliency maps provides an important criterion for the selection of an interpretation scheme and its hyperparameters from the established gradient-based saliency maps in the deep learning literature. Such a theoretical understanding is necessary to avoid the application of the interpretation schemes with a relatively high variance under a limited training set size.

In this paper, we focus on the generalization properties of standard gradient-based interpretation maps, and provide theoretical and numerical results to show that the proper generalization of a NN's gradient-based saliency map could require a larger training set than the standard classification problem focusing only on the accuracy of the prediction model. In other words, the variance of the gradient-based maps with respect to the randomness of training data could be significantly more than the variance of the neural network's predictions.

To support the above statement on the generalization of gradient-based interpretation maps, we prove theoretical bounds on the generalization rate of standard gradient-based saliency maps, including simple and integrated gradients, from training samples to test data. Our generalization bounds indicate the considerable discrepancy between the training and test performance scores of gradient-based interpretation schemes. We compare the shown generalization error bounds with the standard bounds on the generalization error of multi-layer NN classifiers, which suggests a higher statistical complexity for the interpretation of neural nets than for the accuracy of a NN classifier as characterized by Bartlett et al. (2017).

Subsequently, we focus on the SmoothGrad algorithm and show that the Gaussian smoothing in this method can be interpreted as a regularization mechanism controlling the difference between test and training interpretation performance. Our results indicate that the generalization error will decrease linearly with the standard deviation of the SmoothGrad noise, which will reduce the variance of the saliency map at the cost of a higher bias toward a constant interpretation. Therefore, this result would parallel the well-known bias-variance trade-off for norm-based regularization methods in the context of supervised learning.

Finally, we present the results of several numerical experiments demonstrating the effect of the number of training data on the variance of the gradient-based saliency maps. Our empirical findings reveal the significant impact of the size of the training set on the estimated saliency map for unseen test data. We show that standard methods such as simple and integrated gradients are highly susceptible to the samples in the training set. In addition, our results show a lower correlation between gradient-based interpretation maps of two NNs with disjoint training sets than the correlation between the NNs' predicted labels, indicating that an interpretable NN model demands more training data than an accurate NN classifier. Numerically, we show the regularization effect of the SmoothGrad algorithm which manages to properly control the variance of the saliency map on test data. Our numerical results indicate the importance of proper generalization in the visual performance of interpretation methods and support the SmoothGrad approach as a regularized interpretation scheme. Here, we summarize our contributions:

- Highlighting the role of generalization in the performance of deep learning interpretations,
- Proving theoretical generalization bounds for standard gradient-based saliency maps,
- Demonstrating the regularization effect of Gaussian smoothing in SmoothGrad,
- Providing results on interpretations generalization and SmoothGrad regularization.

## 2 RELATED WORK

Standard generalization analysis in deep learning focuses on the consistency of neural nets' predictions across training and test samples. However, neural nets have been shown to memorize random labels and Gaussian pixel inputs (Zhang et al., 2017); to easily overfit dataset biases and labeling errors (Stock & Cisse, 2018; Beyer et al., 2020; Shankar et al., 2020), generating unexplainable predictions and exhibiting weak classification decision boundaries. To debug these faulty predictions, several post-hoc interpretability (Lipton, 2016) methods attempt to explain the outputs via visualizations, counterfactuals and numerical metrics. Unlike multi-modal concept learning methods, such as TCAV (Kim et al., 2018), Concept Bottleneck Models (CBM) (Koh et al., 2020) and Interpretable Basis Decomposition (IBD) (Zhou et al., 2018), post-hoc methods study interpretations as a stand-alone problem independent of the blackbox model training process and setup. In this work,

we choose a different approach by experimenting on gradient-based and feature-based methods, to show that the train-to-test generalization of interpretations depends heavily on training set size.

**Gradient-based Interpretations.** Gradients of the model output with respect to its input is an intuitive way of attributing the prediction to the data representation (Sundararajan et al., 2017). Early attribution techniques generate explanations from the product between simple gradients and features (Baehrens et al., 2010; Simonyan et al., 2013); works such as Guided BackProp (Springenberg et al., 2014), DeConvNet (Zeiler & Fergus, 2014), DeepLift (Shrikumar et al., 2017) and Layer-wise Relevance Propagation (LRP) (Binder et al., 2016) utilize discrete step backpropagation to proportionally attribute class-wise prediction scores to network features. Sundararajan et al. (2017) further improve the reliability of using gradients to weigh feature importance, by proposing integrated gradients to satisfy desirable axioms of sensitivity and implementation invariance.

Gradients also characterise interpretability within and between trained models. Gradient signal-to-noise ratio (GSNR) Liu et al. (2020) uses gradient alignment across different samples to understand representation consistency of a model; Raghu et al. (2021) utilize the norm of network gradients to quantify the amount of discrepancy between the input and prediction. The difference of gradients between 2 networks taken with respect to the same input evaluates how much the networks' predictions disagree. In this work, we experiment on gradient-based feature attribution methods of simple gradients and integrated gradients. We further calculate the norm and distance of networks' gradient interpretations to evaluate prediction consistency and agreement.

**Parameter Space Interpretations.** Beyond gradient-based analysis, the representation similarity of samples between networks and network layers is also an important interpretation metric. Class Activation Mapping (CAM) (Zhou et al., 2016) and the subsequent Grad-CAM (Selvaraju et al., 2017) utilize inherent localization properties of deep NN features to visualize salient regions in images. They project the target class' weights from the output layer back to the convolutional feature maps, using network parameter activations to score the importance of image features for classification. By comparing the CAM interpretations of trained models, we qualitatively assess how consistently do they attend to the same spatial regions. To directly compare between networks and across layers, Centered Kernel Alignment (CKA) (Kornblith et al., 2019) improved upon canonical correlation analysis methods (Raghu et al., 2017; Morcos et al., 2018), by calculating the similarity index between representational matrices. Their results generalize to different kernels, network architectures and layer types, providing us with insight into the similarity between differently trained models, across layers and samples.

**Robustness and Consistency of Interpretations.** Several related papers analyze the fragility and consistency of the standard saliency maps. The related papers (Ghorbani et al., 2019; Dombrowski et al., 2019; Heo et al., 2019; Subramanya et al., 2019) show that standard gradient-based interpretations of neural nets commonly lack robustness to input perturbations, and the manipulated interpretation can transfer across neural net architectures. Levine et al. (2019) present a certifiably robust interpretation scheme by applying sparsification to the SmoothGrad approach. In another related paper, Fel et al. (2022) analyze the consistency and algorithmic stability of standard interpretation methods and measure the sensitivity of interpretation methods to the inclusion of one specific sample in the training set. However, unlike our work, the mentioned works do not focus on the generalization of interpretation methods from training to test data.

## 3 PRELIMINARIES

In this section, we discuss the notation and definitions used throughout the paper and shortly review the gradient-based saliency maps analyzed in the paper.

### 3.1 NOTATION

In the paper, we use notation $\mathbf{X} \in \mathbb{R}^d$ to denote the random feature vector and $Y \in \{1, \ldots, k\}$ to denote the $k$-ary classification label. The deep learning algorithm trains a neural network $f_{\mathbf{w}} \in \mathcal{F}$ where $\mathbf{w}$ represents the vector containing the weights of the neural net function and $\mathcal{F} = \{f_{\mathbf{w}} : \mathbf{w} \in \mathcal{W}\}$ denotes the feasible set of functions including the neural nets with allowed weight vectors in set $\mathcal{W}$. Note that every $f_{\mathbf{w}} : \mathbb{R}^d \to \mathbb{R}^k$ maps the $d$-dimensional input to a $k$-dimensional prediction vector including a real-valued entry for every label.

For training the neural net, we follow the standard empirical risk minimization (ERM) method minimizing the empirical expected loss, measured with loss function $\ell(\hat{y}, y)$ between actual $y$ and predicted $\hat{y}$ labels, over the training set $\{(\mathbf{x}_i, y_i)_{i=1}^n\}$ consisting of $n$ labeled training examples drawn independently from an underlying distribution $P_{\mathbf{X}, Y}$:

$$\min_{\mathbf{w} \in \mathcal{W}} \frac{1}{n} \sum_{i=1}^n \ell\big(f_{\mathbf{w}}(\mathbf{x}_i), y_i\big). \tag{1}$$

We note that the standard generalization analysis in machine learning focuses on the difference between the expected loss values on the training samples and the test samples drawn from the underlying model $P_{\mathbf{X}, Y}$.

### 3.2 GRADIENT-BASED SALIENCY MAPS

In our generalization analysis, we consider standard gradient-based saliency maps as a neural net's interpretation. To define standard saliency maps, we use $f_c(\mathbf{x})$ to denote the real-valued output of the $c$-th neuron at the final layer of neural net $f$. Assuming that $c$ is the assigned label to input $\mathbf{x}$, i.e. the final layer's neuron with the maximum value, we review the definitions of the following standard saliency maps:

1. **Simple Gradient Method**: As defined by Simonyan et al. (2013), the simple gradient is the gradient of the neural net's output at the predicted neuron with respect to the input feature vector:

$$\text{Simple-Grad}(\mathbf{f}_c, \mathbf{x}) := \nabla_{\mathbf{x}} \mathbf{f}_c(\mathbf{x}). \tag{2}$$

2. **Integrated Gradients**: Given a reference vector $\mathbf{x}^0$, the integrated gradients (Sundararajan et al., 2017) calculate the gradient's integral over the line segment connecting the reference point $\mathbf{x}^0$ and a target point $\mathbf{x}$. In practice, the integrated gradient is approximated using $m$ intermediate points between $\mathbf{x}^0$ and $\mathbf{x}$:

$$\text{Int-Grad}(\mathbf{f}_c, \mathbf{x}) := \int_0^1 \nabla_{\mathbf{x}} \mathbf{f}_c\big(\mathbf{x}^0 + \alpha \Delta \mathbf{x}\big) \odot \Delta \mathbf{x} \, d\alpha \approx \frac{\Delta \mathbf{x}}{m} \odot \sum_{i=1}^m \nabla_{\mathbf{x}} \mathbf{f}_c\big(\mathbf{x}^0 + \frac{i}{m} \Delta \mathbf{x}\big). \tag{3}$$

In the above, $\Delta \mathbf{x} = \mathbf{x} - \mathbf{x}^0$ denotes the difference between the reference and target points, and $\odot$ denotes the vector element-wise product.

3. **SmoothGrad**: The SmoothGrad approach (Smilkov et al., 2017) applies Gaussian smoothing to the gradient-based interpretation, and calculates the average gradient with an isotropic Gaussian distribution centered at the target data point $\mathbf{x}$. Specifically, we define Gaussian vector $\mathbf{Z} \sim \mathcal{N}(\mathbf{0}, \sigma^2 I)$ and define SmoothGrad as

$$\text{Smooth-Grad}(\mathbf{f}_c, \mathbf{x}) := \mathbb{E}_{\mathbf{Z} \sim \mathcal{N}(\mathbf{0}, \sigma^2 I)}\big[\nabla f_c(\mathbf{x} + \mathbf{Z})\big] \approx \frac{1}{m} \sum_{i=1}^m \nabla f_c(\mathbf{x} + \mathbf{z}_i), \tag{4}$$

where $\mathbf{z}_1, \ldots, \mathbf{z}_m \sim \mathcal{N}(\mathbf{0}, \sigma^2 I)$ are independent observations of the Gaussian noise used to approximate the SmoothGrad expectation.

## 4 GENERALIZATION IN INTERPRETATION TASKS

Generalization from training examples to test data is a crucial factor behind the success of every learning algorithm. In the case of interpretation methods, we note that the trained neural net $f_{\mathbf{w}} \in \mathcal{F}$ is learned using the training data, and hence the learned function will be different from the optimal neural net minimizing the expected loss over the underlying distribution of test data $P_{\mathbf{X}, Y}$. In our discussion, we use $f^*$ to denote an optimal classifier in $\mathcal{F}$ in terms of the achieved performance on the underlying distribution of data, i.e.

$$f^* \in \operatorname*{argmin}_{f \in \mathcal{F}} \mathbb{E}_{(\mathbf{X}, Y) \sim P}\big[\ell\big(f(\mathbf{X}), Y\big)\big]. \tag{5}$$

**Remark 1.** *In the following discussion, we suppose a unique minimizer $f^*$ to the above risk minimization problem. Note that if this assumption does not hold, we define $f^*(\mathbf{x}) = \mathbb{E}_{f \sim \text{Unif}(F^*)}[f(\mathbf{x})]$*

*as the expectation of the classifier's output according to the uniform distribution on the set $F^*$ of optimal solutions to the above population risk minimization problem. In addition, we note that the following definitions and Theorems 1, 2 will similarly hold under an alternative definition $f^*(\mathbf{x}) = \mathbb{E}_{S,A}[f_{A(S)}(\mathbf{x})]$ where the expected value is taken over the randomness of the size-$n$ training set $S$ independently drawn from $P_{\mathbf{X},Y}$ and stochastic learning algorithm $A$ learning weights $A(S)$ to obtain the classifier neural net $f_{A(S)}$.*

While we, as the learner, do not know the underlying distribution $P_{\mathbf{X},Y}$ and therefore the optimal $f^*$, we can still define the loss of an interpretation scheme $I(\cdot)$ at an input $\mathbf{x}$ as the norm difference between $I$'s output for a given classifier $f$ and the optimal $f^*$, that is

$$\text{Loss}_I(f, \mathbf{x}) := \big\| I(f, \mathbf{x}) - I(f^*, \mathbf{x}) \big\|_2, \tag{6}$$

where $\|\cdot\|_2$ denotes the $L_2$-norm of an input vector. Here we define the interpretation vector $I(f, \mathbf{x})$ when we choose class $c = y$ for the actual label $y$ of sample $\mathbf{x}$. Also, note that the above definition uses $I(f^*, \mathbf{x})$ as the underlying interpretation which the learner aims to estimate from training data.

**Definition 1.** *For a classifier function $f$ and training set $\{(\mathbf{x}_i, y_i)_{i=1}^n\}$, we define the interpretation training loss $\widehat{\mathcal{L}}(f)$ as the expected interpretation loss on training data:*

$$\widehat{\mathcal{L}}(f) := \frac{1}{n} \sum_{i=1}^n \text{Loss}_I(f, \mathbf{x}_i).$$

*Also, we define the interpretation test loss $\mathcal{L}(f)$ as the expected interpretation loss on the underlying distribution of test data $P_{\mathbf{X}}$:*

$$\mathcal{L}(f) := \mathbb{E}_{\mathbf{X} \sim P_{\mathbf{X}}} \big[ \text{Loss}_I(f, \mathbf{X}) \big].$$

*Finally, we define the interpretation generalization error as the difference between the interpretation training and test loss values:*

$$\epsilon_{\text{gen}}(f) := \mathcal{L}(f) - \widehat{\mathcal{L}}(f).$$

Based on the above definition, a necessary condition to have a controlled variance of the gradient-based interpretation map is a bounded interpretation generalization error. In the next section, we present theoretical bounds on the interpretation generalization error of neural network classifiers, to compare the generalization rates for several standard gradient-based interpretation maps.

## 5 THEORETICAL BOUNDS ON INTERPRETATION GENERALIZATION ERROR

In this section, we theoretically analyze the interpretation generalization error of neural networks. Here we suppose that the neural net function $f_{\mathbf{w}} : \mathbb{R}^d \to \mathbb{R}^k$ has the following format:

$$f_{\mathbf{w}}(\mathbf{x}) = W_L \phi_{L-1} \big( W_{L-1} \phi_{L-2}(\cdots W_2 \phi_1(W_1 \mathbf{x})) \cdot \big). \tag{7}$$

Here the vector $\mathbf{w}$ concatenates the entries of the $L$ layers' weight matrices $W_1, \ldots, W_L$. Also, $\phi_i : \mathbb{R} \to \mathbb{R}$ represents the activation function at layer $i$.

Our first theorem concerns the interpretation generalization performance of the simple gradient and integrated gradients. This result demonstrates that the generalization of these gradient-based interpretation schemes could require a larger training set than the standard deep learning classification problem. Specifically, this theorem extends the generalization analysis in Bartlett et al. (2017) to the gradient-based interpretation of neural networks. In the following, we use $\|\cdot\|_2$ to denote a matrix's spectral norm, i.e. its largest singular value, and also $\|\cdot\|_{2,1}$ denotes the $L_{2,1}$-group norm of a matrix, i.e. the summation of the $L_2$-norms of the matrix's rows.

**Theorem 1.** *Suppose that the neural net classifier in equation 7 has an $\gamma_i$-Lipschitz and $\gamma_i$-smooth activation function satisfying $\forall z \in \mathbb{R} : \max\{|\phi_i'(z)|, |\phi_i''(z)|\} \le \gamma_i$. We assume that the interpretation loss is upper-bounded by constant $c$ and the training data matrix $\mathbf{X}_{n \times d}$ is norm-bounded as $\|\mathbf{X}\|_2 \le B$ with probability 1. Also, we use $D$ to denote the maximum number of rows and columns in $f_{\mathbf{w}}$'s weight matrices. Then, for every $\omega > 0$, with probability at least $1 - \omega$ this generalization error bound will hold for both the simple gradient method and integrated gradients of every $f_{\mathbf{w}}$:*

$$\epsilon_{\text{gen}}(f_{\mathbf{w}}) \le \mathcal{O}\Big( c \sqrt{\frac{\log(1/\omega)}{n}} + \frac{B R_{\mathbf{w}} \log(n) \log(D)}{n} \Big).$$

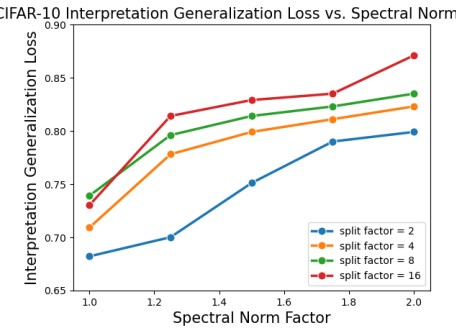 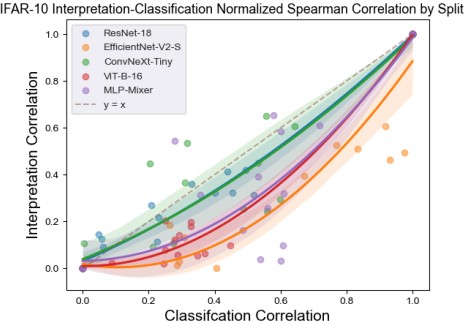

(a) Interpretation generalization loss vs. spectral norm factor.

(b) CIFAR-10 interpretation-classification correlation scores.

Figure 1: We find that networks with pre-training and spectrally-normalized networks using smaller (stricter) norm factors exhibit lower interpretation generalization loss.

*Here $R_{\mathbf{w}} := \big(\sum_{i=1}^{L}\prod_{j=1}^{i}\gamma_j\|\mathbf{W}_j\|_2\big)\big(\prod_{i=1}^{L}\gamma_i\|\mathbf{W}_i\|_2\big) \times \big(\sum_{i=1}^{L}\frac{\|W_i\|_{2,1}^{2/3}}{\|W_i\|_2^{2/3}}\big)^{3/2}$ denotes the interpretation capacity of the neural net.*

Comparing the generalization bound for the simple and integrated gradients interpretation to the generalization bound in Bartlett et al. (2017) for the standard supervised learning task, we notice an order-wise $\mathcal{O}\big(\sum_{i=1}^{L}\prod_{j=1}^{i}\gamma_j\|\mathbf{W}_j\|_2\big)$ greater generalization error for gradient-based interpretation schemes. This additional term indicates the extra cost of generalization for the simple and integrated gradients-based interpretation scheme. The generalization comparison with deep supervised learning could be also performed using the results of (Neyshabur et al., 2018; Galanti et al., 2023). Next, we state the generalization bound for the SmoothGrad approach.

**Theorem 2.** *Suppose that the neural net classifier in equation 7 has an $\gamma_i$-Lipschitz activation function satisfying $\forall z \in \mathbb{R}: |\phi_i'(z)| \leq \gamma_i$. We assume that the interpretation loss is upper-bounded by constant $c$ and the training data matrix $\mathbf{X}_{n \times d}$ is norm-bounded as $\|\mathbf{X}\|_2 \leq B$ with probability 1. Then, for every $\omega > 0$, with probability at least $1 - \omega$ the following generalization error bound will hold for the SmoothGrad interpretation of every $f_{\mathbf{w}}$ with standard deviation $\sigma > 0$:*

$$\epsilon_{\text{gen}}(f_{\mathbf{w}}) \leq \mathcal{O}\Big(c\sqrt{\frac{\log(1/\omega)}{n}} + \frac{BL_{\mathbf{w}}\log(n)\log(D)\sqrt{d}}{n\sigma}\Big),$$

*where $L_{\mathbf{w}} := \prod_{i=1}^{L}\gamma_i\|\mathbf{W}_i\|_2\big(\sum_{i=1}^{L}\frac{\|W_i\|_{2,1}^{2/3}}{\|W_i\|_2^{2/3}}\big)^{3/2}$ denotes the spectral capacity of the neural net.*

Note that Theorem 2's bound is only by a multiplicative factor $\frac{\sqrt{d}}{\sigma}$ different from the generalization bound in the standard deep supervised learning problem (Bartlett et al., 2017). Therefore, the theorem suggests that Gaussian smoothing can be interpreted as a regularization of the simple gradient approach to improve its generalization behavior. The SmoothGrad interpretation algorithm could gain a better generalization performance by increasing the standard deviation, while the training performance could drop because of the additional noise.

## 6 NUMERICAL EXPERIMENTS

### 6.1 EXPERIMENTAL DETAILS

**Datasets.** We numerically study the generalization and visual consistency of interpretation methods on the standard CIFAR-10 (Krizhevsky et al., 2009) and the larger scale TinyImageNet (Le & Yang, 2015) and Caltech-256 (Griffin et al., 2022) datasets. TinyImageNet dataset is a downsampled subset of ImageNet (Deng et al., 2009) and comprises 200 object categories with 500 training images and 50 validation images for each class. Caltech-256 contains 256 object categories totaling 30,607 high-resolution images. We note that since our experiments would require us to train from scratch a

Table 1: Rank correlation coefficient and saliency pixel intersection on the test set, for the interpretations of neural nets trained with a training set split factor of sf $= 2, 4, 8, 16$.

| | | ResNet-18/50 | | ViT-B-16 | | MLP-Mixer | |
|---|---|---|---|---|---|---|---|
| | sf | Rank C ↑ | Px % ↑ | Rank C ↑ | Px % ↑ | Rank C ↑ | Px % ↑ |
| CIFAR-10 | 2 | **.37** $\pm$ .02 | **28.0** $\pm$ 0.9 | **.31** $\pm$ .02 | **23.7** $\pm$ 1.6 | **.39** $\pm$ .01 | **34.1** $\pm$ 1.3 |
| | 4 | .33 $\pm$ .01 | 26.8 $\pm$ 1.0 | .25 $\pm$ .02 | 18.4 $\pm$ 1.5 | .38 $\pm$ .01 | 33.6 $\pm$ 1.0 |
| | 8 | .31 $\pm$ .01 | 25.3 $\pm$ 0.7 | .25 $\pm$ .02 | 17.7 $\pm$ 1.3 | .36 $\pm$ .01 | 32.7 $\pm$ 0.9 |
| | 16 | .28 $\pm$ .01 | 23.8 $\pm$ 0.5 | .23 $\pm$ .02 | 15.4 $\pm$ 1.4 | .34 $\pm$ .01 | 28.4 $\pm$ 0.9 |
| Caltech-256 | 2 | **.31** $\pm$ .01 | **3.3** $\pm$ 0.1 | **.21** $\pm$ .05 | **3.8** $\pm$ 1.4 | **.31** $\pm$ .01 | **4.3** $\pm$ 0.3 |
| | 4 | .30 $\pm$ .02 | 2.2 $\pm$ 0.3 | .18 $\pm$ .05 | 1.7 $\pm$ 0.5 | .21 $\pm$ .05 | 1.5 $\pm$ 0.4 |
| | 8 | .27 $\pm$ .04 | 1.7 $\pm$ 0.6 | .17 $\pm$ .02 | 0.7 $\pm$ 0.6 | .18 $\pm$ .03 | 1.2 $\pm$ 0.7 |
| | 16 | .24 $\pm$ .03 | 0.1 $\pm$ 0.4 | .15 $\pm$ .03 | 0.5 $\pm$ 0.6 | .14 $\pm$ .02 | 0.9 $\pm$ 0.7 |
| Tiny-ImageNet | 2 | **.12** $\pm$ .02 | **23.8** $\pm$ 0.9 | **.11** $\pm$ .01 | **20.3** $\pm$ 0.2 | **.11** $\pm$ .01 | **26.3** $\pm$ 0.3 |
| | 4 | .10 $\pm$ .03 | 23.7 $\pm$ 0.5 | .10 $\pm$ .01 | 20.1 $\pm$ 1.1 | .06 $\pm$ .03 | 22.4 $\pm$ 0.4 |
| | 8 | .06 $\pm$ .03 | 21.4 $\pm$ 0.5 | .05 $\pm$ .02 | 18.8 $\pm$ 1.0 | .03 $\pm$ .02 | 18.4 $\pm$ 1.0 |
| | 16 | .03 $\pm$ .03 | 20.0 $\pm$ 0.8 | .05 $\pm$ .01 | 18.3 $\pm$ 0.5 | .03 $\pm$ .01 | 18.3 $\pm$ 0.9 |

multitude networks on different subset levels for each dataset, it was infeasible to directly experiment on the large-scale ImageNet (Deng et al., 2009) dataset. Instead, to validate the message that the generalization of interpretations requires more data, we utilize the large-scale ImageNet dataset for pre-training via off-the-shelf weights.

**Neural network architectures.** To validate our hypotheses, we experiment on a diverse set of computer vision architectures. We report numerical results for the following convolutional neural networks: ConvNeXt-Tiny (Liu et al., 2022), EfficientNet-V2-S (Tan & Le, 2021), ResNet (He et al., 2016) (we trained ResNet-50 on Caltech-256 and trained ResNet-18 on TinyImageNet, CIFAR-10); for the ViT-B-16 Vision Transformer model proposed by Beyer et al. (2022); and for the multi-layer perceptron model of MLP-Mixer (Tolstikhin et al., 2021).

**Experiment design.** To evaluate the effect of training set size on interpretation generalization, we consider split factors of sf $= 2, 4, 8, 16$, each corresponding to training with $50\%, 25\%, 12.5\%, 6.25\%$ of available training data. We train a neural net for every data subset for 200 epochs. To further improve the interpretation generalization, we allow models to train on "more data" by using pre-trained ImageNet weights, then fine-tuning for 50 epochs.

### 6.2 VERIFYING THE GENERALIZATION GAP

In Figure 1b, we show that network interpretation performance suffer more than network classification performance, under the effect of training set scale and overlap. On test set data, we plot the normalized Spearman correlation of network interpretations against softmax predictions. As $sf$ increases from 2 to 16 and models are trained with smaller, more disjoint training sets, the rank correlation of test set interpretations drop more acutely than that of network predictions. Results of other datasets are in the Appendix.

Furthermore, we visualize the interpretation generalization gap in Fig. 2, by varying the number of training samples from 6.25% of the training set, to pre-training on ImageNet and fine-tuning on 50% of the train set. As the number of training samples increased, GradCAM (Selvaraju et al., 2017) interpretations became more similar between pairs of models, as seen from how "Pretrain" model pairs have near-perfect saliency map agreement across datasets. For models that are optimized with more training samples, this localization ability transfers successfully to unseen test data, verifying that more training samples are required for interpretations to agree across models and generalize across train and test sets.

### 6.3 GRADIENT-BASED INTERPRETATIONS

In Figure 3, via qualitative experiments on Caltech-256 (Griffin et al., 2022) with the simple gradient (Simonyan et al., 2013), SmoothGrad (Smilkov et al., 2017), integrated gradients (Sundararajan et al., 2017) and DeepLift (Shrikumar et al., 2017), we show that "Pretrain" models outperform

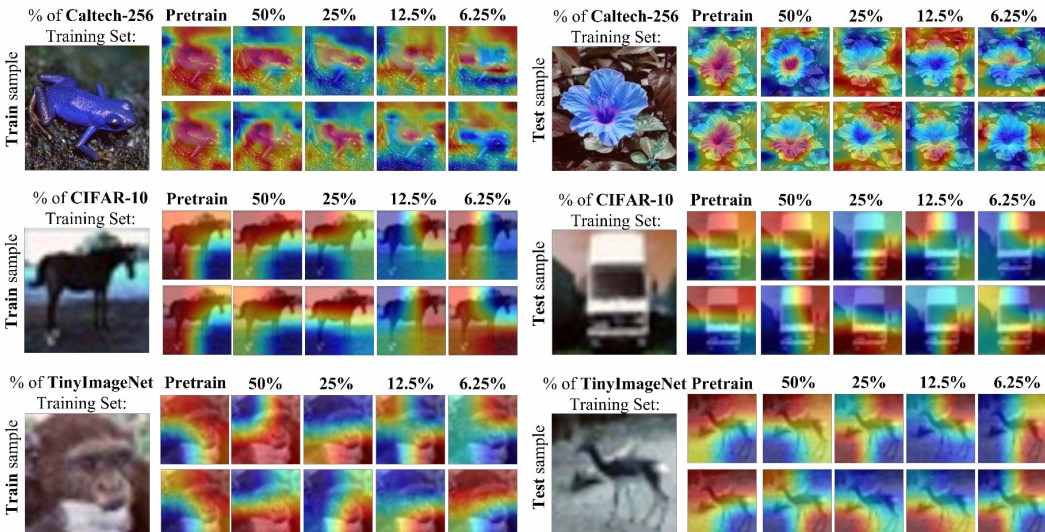

Figure 2: Grad-CAM comparisons with ConvNeXt-Tiny. We experiment with models trained on disjoint (thus independent) splits of the training set with $sf = 2, 4, 8, 16$. As we increase the number of training samples from 1) 6.25% ($sf = 16$) of the training set, to 2) using 50% of the training set, then to 3) pre-training on ImageNet plus fine-tuning with 50% training data, we observe that model pairs generate increasingly consistent interpretations.

6.25% models in terms of visual fidelity, localization meaningfulness and generalization ability to test samples. We present further numerical evidence by assessing the generalization gap in *integrated gradients* (Sundararajan et al., 2017). We vary the dataset split factor from 2 ($\frac{1}{2}$ of train set), 4, 8 to 16 ($\frac{1}{16}$ of the train set) and generated mass-centered perturbations with the attacker network for the source network. The intuition behind this technique is that if the networks have similar gradient interpretations, then the perturbations generated by the attacker would have negligible effect on the source networks' saliency map outputs. In Table 1, we compare the *a) rank correlation of saliency maps*, the Spearman rank correlation coefficient between saliency maps of original and perturbed images; *b) top-100 salient pixel intersection %*, indicating the percentage of overlap between the top-100 most salient pixels, which are used for classifying the original and perturbed images. Our comparison shows a consistent improvement of the metrics by increasing the sample size.

## 6.4    IMPROVING GENERALIZATION VIA SPECTRAL NORMALIZATION AND SMOOTHGRAD

Motivated by our theoretical results in Theorem 1, which suggest the application of *spectral normalization* in closing the interpretation generalization gap, we numerically validate this in Figure 1a. By plotting the interpretation generalization loss against the spectral norm factor of spectrally-normalized neural nets, we verify that a lower (stricter) normalization factor leads to lower generalization loss; this demonstrates the practical implication of Theorem 1.

To further improve generalization performance, we emphasize Theorem 2, which reveals the regularization effect of Gaussian smoothing in SmoothGrad to decrease the generalization gap. This is a non-trivial result explaining why SmoothGrad substantially improves SimpleGrad and Integrated-gradients; we conduct experiments comparing these methods. Our goals are to first quantify the within-model discrepancy (mis-attribution) between the input and output and second to evaluate how the cross-network gradient-based interpretations increasingly disagree with fewer training samples. We subsequently compute the mean $L_2$-norm difference of the interpretation vectors for networks with disjoint training sets of the same size. A larger norm difference indicates a greater discrepancy between the interpretations and worse generalization.

We report results averaging over $m = 1, 5, 20, 50$ Gaussian noise vectors for the estimation of the SmoothGrad interpretation, with Gaussian perturbation standard deviation $\sigma$ chosen from the set $\{0, 0.1, 0.2, 0.5, 1.0, 2.0, 5.0\}$. We observe that increasing the number of randomly-perturbed samples with Gaussian noise has a gradient smoothing effect. Also, as visualized in Appendix

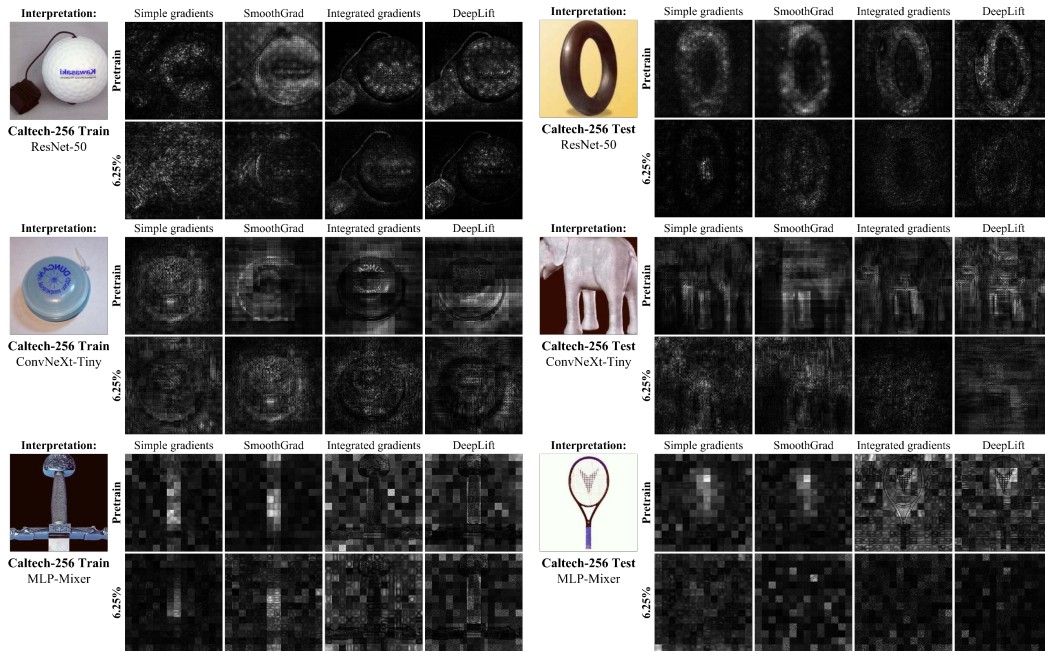

Figure 3: Different gradient-based interpretation methods tested on Caltech-256. We compare the fidelity, localization meaningfulness and train-test generalization abilities of interpretations, for Pretrain and the 6.25% settings. The generalization and performance gaps widen for interpretations generated by models trained on smaller, disjoint training sets. Full results are in the Appendix.

Figures 8-15, increasing the noise standard deviation improves Gaussian smoothing power, with effects of increasing the interpretation agreement and reducing the generalization gap. Comparing the simple gradient (marked by "no $\sigma$" in the legends) and SmoothGrad methods' results, we observe that the Gaussian smoothing in SmoothGrad improves cross-network interpretation agreement and hence the generalization of the gradient-based saliency map. This observation is consistent with our theoretical analysis, evidencing the regularization role of Gaussian smoothing in SmoothGrad.

## 7 CONCLUSION

In this paper, we highlight the role of proper generalization from training samples to unseen test data in the success of deep learning-based interpretation methods. On the theory side, we prove generalization error bounds to show the higher sample complexity of learning interpretable neural net classifiers, and further discuss the regularization effect of Gaussian smoothing in the SmoothGrad approach. On the empirical side, our numerical results also demonstrate the influence of the training set size on the generalization of gradient-based interpretation methods to test samples. To further expand the analysis, an interesting future direction is to explore other regularization schemes and their effect on the generalization of interpretation methods. Such a study can be performed for popular deep learning regularization schemes such as batch normalization and dropout. Furthermore, the extensions of our generalization study to mask-based and perturbation-based explanation tools could improve the understanding of the effect of adversarial schemes on the generalization properties of the interpretability of neural networks. We note that our developed generalization framework is relatively general and potentially applicable for studying the discussed future directions.

## 8 REPRODUCIBILITY STATEMENT

To ensure reproducibility, we have attached our source code to the supplement. We also include details on datasets, network architectures and experiment design in Section 6. We note that our setups–including but not limited to the choice of benchmarks, baselines, metrics–are consistent with existing literature and key references such as Ghorbani et al. (2019) and Smilkov et al. (2017).

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
