# OpenReview forum: "On the Generalization of Gradient-based Neural Network Interpretations"
_ICLR.cc/2024/Conference — Submitted to ICLR 2024_

### Official Review · Reviewer_Cg7V · 2023-10-28

**Soundness:** 2 fair
**Presentation:** 3 good
**Contribution:** 1 poor
**Rating:** 3
**Confidence:** 3

**Summary:**

**Update:** I would like to thank the authors for their response, and I have read all the reviews and the corresponding responses.
I decide to keep my original score as my main concerns remain. Specifically, I remain unconvinced about the significance of studying the generalization of interpretation methods. Consider the reference classifier in your revised version, namely $f^*(x)=\mathbb{E}\_{f\sim\mathrm{Unif}(F^*)}f(x)$, the question arises: is $\mathbb{E}\_{f\sim\mathrm{Unif}(F^*)}||I(f,x)-I(f^*,x)||_2$ small? As mentioned in my previous question 2, if the optimal $f$ set can have large variance/standard deviation, why does it matter if the saliency maps are sensitive to the training data?
___


This paper investigates the generalization of gradient-based interpretation methods in deep learning. The authors demonstrate the significant influence of the sample size on the interpretation, such as saliency maps, in deep neural networks. Then they derive two generalization bounds for common gradient-based interpretation techniques by using the analysis presented in Bartlett et al. (2017). Notably, for SmoothGrad, they show that the generalization error of interpretation has a linear decrease with the standard deviation of the SmoothGrad noise. The paper complements these findings with numerical results demonstrating the impact of training sample size on interpretation outcomes.

**Strengths:**

The paper is well-written and investigates a topic that has not been explored before, namely the generalization of gradient-based interpretation.

**Weaknesses:**

My primary concern lies in understanding the importance of investigating the generalization aspect of interpretation methods, and the validation of the generalization definition in this paper. Additionally, the theoretical contributions in this paper are somewhat restricted, as a substantial portion of the analysis is derived from Bartlett et al. (2017), with the key distinction being the definition of "loss" and "error".

For more detailed questions, please refer below.

**Questions:**

1. Regarding the definition of $f^*$ in Eq.(5):
Firstly, there is no guarantee that you only have one such optimal classifier. In the case where Eq.(5) returns a set of $f^*$, doesn't the definition of loss in Eq.(6) become problematic? While all the $f^*$ will yield the same testing performance, they do not necessarily produce the same output for the interpretation method. In other words, $\mathrm{Loss}_I(f,x)$ is also a function of $f^*$, denoted as $\mathrm{Loss}_I(f^*,f,x)$. This differs significantly from studying standard generalization error. Therefore, the generalization error defined in Eq.(6) can vary for the same $f$ and $x$.

2. Another question arises in this context. Considering that there could be multiple $f^*$ with identical testing performance, and they may not produce the same saliency maps, why does it matter if the saliency maps are influenced by the training data? Is there any guarantee that different $f^*\in\mathcal{F}$ with different weight parameters will return identical outputs from the interpolation methods (for the same $x$)?

3. Let's assume there is only one such $f^*$. According to Theorem 1, the generalization bound implies that interpretation demands a larger training set compared to the standard classification problem. An essential mystery in the success of deep learning is that overparameterized neural networks can generalize well without needing more data than the number of parameters. If Simple Gradient Method and Integrated Gradient prove to be unreliable with the same amount of data, does this imply that they are ineffective for interpreting deep neural networks?

4. What is the fundamental implication of establishing generalization bounds for gradient-based interpretation methods when we cannot ensure good performance on the training data? In the context of standard generalization error, we can say an upper bound is provided to guide the minimization of empirical risk while controlling key quantities in the bound.  However, for interpretation methods, let $\hat{f}^*$ be the empirical minimizer for a given training dataset, even if standard training leads to $f_w\to\hat{f}^*$, $\mathrm{Loss}_I(\hat{f}^*,x)=||I(\hat{f}^*,x)-I(f^*,x)||_2$ can still be large. In other words, we lack clarity on whether interpretation methods might overfit to the training data (in the sense defined in Eq.(5)). In light of this uncertainty, discussing regularization is not feasible at this point.

---

> ### Author Response · Authors · 2023-11-22
> **Authors' Response to Reviewer Cg7V**
>
> We thank Reviewer Cg7V for his/her time and feedback. Here is our response to the comments and questions in the review:
>
> **1- “The importance of investigating the generalization aspect of interpretation methods”**
>
> **Re:** Please refer to our general response to comment 1.
>
> **2- The role of** $f^*$ **in our definition of interpretation loss**
>
> **Re:**  Please refer to our general response to comment 2.
>
> **3- "There is no guarantee that you only have one such optimal classifier."**
>
> **Re:** We note that if the minimizer to the population risk function is not unique, we could define  $f^*(x)=E_{f\sim \mathrm{Unif}(F^*)} [f(x)]$ as the expectation under the uniform distribution on the set $F^*$ of optimal population solutions, and our generalization guarantees for Theorems 1,2 will hold for this choice of $f^*$.
>
> Furthermore, as we discussed in the general response to comment 2, $I(f^*,x)$ in our definition of interpretation loss can be changed to $E_{S,A}[I(f_{A(S)},x)]$ where we take the expected value of  the interpretation map over the randomness of training set $S$ of size $n$ and a stochastic learning algorithm $A$. In that case, we have a unique expected value which can also be used for the analysis. Please note that the same Theorems 1,2 will also hold for the choice of reference map $I(f^*,x) = E_{S,A}[I(f_{A(S)},x)]$.
>
>
> **4-“If Simple Gradient Method and Integrated Gradient prove to be unreliable with the same amount of data, does this imply that they are ineffective for interpreting deep neural networks?”**
>
>  **Re:** This question is precisely the motivation of our generalization study, which highlights the importance of the generalization error (or variance) of an interpretation map with respect to the randomness of training data in the selection of an explanation method and its hyperparameters. For example, if SmoothGrad has a significantly lower generalization error in a learning setting, should a practitioner prefer SmoothGrad over SimpleGrad? As we argued in our general response to comment 1, we believe the answer to this question is yes and our work aims to better understand the generalizability of gradient-based maps.

---

### Official Review · Reviewer_DbhP · 2023-11-01

**Soundness:** 3 good
**Presentation:** 3 good
**Contribution:** 3 good
**Rating:** 6
**Confidence:** 4

**Summary:**

The work derives generalization bounds incorporating gradient based interpretations, which yield non-trivial results. It shows that the generalization of interpretations requires more training, and show that it can be improved with spectral normalization.

**Strengths:**

The paper is interesting, and the bound is both non-trivial and important. It shows that generalization bounds may incorporate intuitive signals for the human observer. Further, it yields more into how neural network interpretations work. The empirical experiments match the theoretical results.

**Weaknesses:**

The overall paper is good other than minor comments on the presentation (see Questions)

**Questions:**

Figure 1: the text and legend can be enlarged/improved. How were the lines produced? what exactly is shown in Figure 1b? What does each point represent? It seems like the lines don't represent the data. Have the authors considered different seeds for each network to add more points to the graph?

** Possible missing references:**

[1] Galanti, T., Galanti, L., & Ben-Shaul, I. (2023). Comparative Generalization Bounds for Deep Neural Networks. Transactions on Machine Learning Research, (ISSN 2835-8856).

---

> ### Author Response · Authors · 2023-11-22
> **Authors' Response to Reviewer Dbhp**
>
> We thank Reviewer Dbhp for his/her time and feedback on our work. In the following, we respond to the comments and questions of the reviewer.
>
> **1- Figure 1b**
>
> **Re: In Figure 1.b, we plot the Spearman rank correlation between the two interpretation maps of the networks $f_{S_1},f_{S_2}$ trained under disjoint training sets $S_1,S_2$ in the y-axis vs. the (Pearson) correlation coefficient between the argmax (i.e. prediction) of the softmax layer (i.e. the fraction of samples with the same argmax prediction by $f_{S_1},f_{S_2}$) in the x-axis. For each neural network architecture, represented by a different color in the figure, we show the points corresponding to split factors $\frac{1}{2},\frac{1}{4},\frac{1}{8},\frac{1}{16}$. We have made this point more clear in the revision. The training of all the networks in the experiments have been initialized at different random initializations to simulate the full randomness of the stochastic optimization algorithm for training the neural nets.
>
> **2- Missing Reference**
>
> **Re:** We thank the reviewer for mentioning the missing reference, which we have discussed in the revision.

---

### Official Review · Reviewer_Yqqc · 2023-11-03

**Soundness:** 3 good
**Presentation:** 2 fair
**Contribution:** 2 fair
**Rating:** 3
**Confidence:** 4

**Summary:**

The paper studies the problem of generalization of gradient-based saliency maps of deep networks. Theoretical bounds are shown and experiments are carried out to validate the results.

**Strengths:**

This paper is well written upto the experiments section. After that I find it difficult to comprehend the presentation. Details in Questions.

**Weaknesses:**

I find the main problem the paper is trying to address somewhat contrived. Firstly, the usual motivation for post-hoc explanation of deep networks is to explain the prediction of a **given** network (trained or otherwise) on a **given** sample, as we get insights as to why a particular decision was made. From this perspective, I do not see the motivation to study how well the input gradients will generalize from train set to test set (in expectation). Why are we interested in knowing the MSE loss between our current network an the optimal network as defined in eq(6).

Continuing on my first point, the authors further claim "the generalization condition is necessary for a proper interpretation result on test samples, it is still not sufficient for a satisfactory interpretation performance". I fail to appreciate this statement since it is not clear satisfactory interpretation (as defined by the authors) is the gradient-based saliency map generated for our optimal f* that minimizes the population loss. In my opinion, this is of little interest from the perspective of understand why a given network made a particular decision. I invite the authors to convince me otherwise.

Lastly, post-hoc explanations have been criticized for some time now in the community due to their reliability in explaining deep network presentations [1,2,3] (some of the references analyze the saliency map techniques evaluated in this work). This means that even if I had the **exact** same saliency map as the ground truth f*, methods like integrated gradients and simple gradients are simply unreliable in explaining what the deep network is doing.

Given, the above three arguments I fail to appreciate the utility of studying the generalization of saliency maps for deep networks.

Lastly, I find the experiment section not clearly written. I expand on this in specific questions below.

1. Adebayo, J., Gilmer, J., Muelly, M., Goodfellow, I., Hardt, M., & Kim, B. (2018). Sanity checks for saliency maps. Advances in neural information processing systems, 31.
2. Shah, H., Jain, P., & Netrapalli, P. (2021). Do input gradients highlight discriminative features?. Advances in Neural Information Processing Systems, 34, 2046-2059.
3. Adebayo, Julius, et al. "Post hoc explanations may be ineffective for detecting unknown spurious correlation." International conference on learning representations. 2021.

**Questions:**

1. Figure 2 is not clear. The caption says "we observe that model pairs generate increasingly consistent interpretations." It is not clear o me what is being compared here for consistent interpretation? Since f* (the optimal classifier) is never accessible to us. What is the baseline here then?

2. The following statement is unclear "We train a neural net for every data subset
for 200 epochs. To further improve the interpretation generalization, we allow models to train on
“more data” by using pre-trained ImageNet weights, then fine-tuning for 50 epochs." Why are we training for 200 epochs and then taking Pretrained weights and fine-tuning for 50 more epochs? Do we start with the fine tuned weights and train for 250 epochs. This should be made clear.

3. The following is unclear "On test set data, we plot
the normalized Spearman correlation of network interpretations against softmax predictions." What exactly is the equations for this computation? Is the softmax predictions the argmax of the softmax or the entire k dimensional softmax scores? What are the two quantities among which the correlation is computed?

---

> ### Author Response · Authors · 2023-11-22
> **Authors' Response to Reviewer Yqqc**
>
> We thank Reviewer Yqqc for his/her time and feedback. The following is our response to the reviewer’s comments:
>
> **1- The motivation to study how well the input gradients will generalize from train set to test set**
>
> **Re:** Please refer to our general response to comment 1.
>
> **2-“it is not clear satisfactory interpretation (as defined by the authors) is the gradient-based saliency map generated for our optimal** $f^*$ **that minimizes the population loss”**
>
> **Re:** Please refer to our general response to comment 2.
>
> **3- “This means that even if I had the exact same saliency map as the ground truth** $f^*$, **methods like integrated gradients and simple gradients are simply unreliable in explaining what the deep network is doing.”**
>
> **Re**: We would like to clarify that our generalization analysis does not focus on finding the interpretation map under ground-truth solution $f^*$. As argued in the general response to the previous comment, we only attempt to measure and control the variance of the interpretation map with respect to the randomness of training data used for training the neural network. Also, our analysis reveals the application of Gaussian smoothing in SmoothGrad to control the variance of the interpretation map.
>
>
> **4- (Figure 2) "Since** $f^*$ **(the optimal classifier) is never accessible to us. What is the baseline here then?"**
>
> **Re:** As we have described in the paragraph titled "Experimental design" on page 7, we have split the original training set into disjoint subsets with split factors $\frac{1}{2},\frac{1}{4},\frac{1}{8},\frac{1}{16}$ and Figure 2 shows how the Grad-Cam maps for a group of test data will look for two neural nets trained using those disjoint (thus independent) training sets. In this experiment, we do not consider any population solution $f^*$ and only display the consistency of the interpretation maps between neural nets trained with independent training data, to show how the consistency could vary with the size of the training set.
>
> **5- “Why are we training for 200 epochs and then taking pretrained weights and fine-tuning for 50 more epochs?”**
>
> **Re:** The size of the datasets we used in our numerical analysis are all bounded by 100,000. Therefore, even for the maximum split factor $\frac{1}{2}$ we could compare the consistency of interpretation maps for neural nets trained using maximum 50,000 training data. To simulate a setting where the two neural nets are trained on more samples, we follow the idea of initializing the neural net at a pretrained network on ImageNet (trained on more than 1 million samples) and applying 50 epochs of training on the samples in the $\frac{1}{2}$-splits of the original training sets. The numerical results in Figure 2 shows that this idea significantly improves the consistency between the two networks’ interpretation maps, supporting our hypothesis that using more training data could lead to less variance in the interpretation maps.
>
>
> **6- “Is the softmax predictions the argmax of the softmax or the entire k dimensional softmax scores?”**
>
> **Re:** In Figure 1.b, we plot the Spearman rank correlation between the two interpretation maps of the networks $f_{S_1},f_{S_2}$ in the y-axis versus the (Pearson) correlation coefficient between the argmax (i.e. prediction) of the softmax layer (i.e. the fraction of samples with the same argmax prediction by $f_{S_1},f_{S_2}$) in the x-axis. We have clarified this point in the revision.

---

### Author Response · Authors · 2023-11-22
**Authors' General Response**

We thank the reviewers for their time and feedback on our work. Here, we respond to the common comments raised by Reviewers Yqqc and Cg7V. We provide our responses to the other questions and comments under each review.

**1- Motivation of our study of the generalization of interpretation maps**

**Re:**  In the deep learning literature, there are several widely-known methods for generating interpretation maps for neural net classifiers, including SimpleGrad, Integrated Gradients, DeepLIFT, SmoothGrad, Sparisified SmoothGrad, etc. Also, the application of these methods requires the selection of some hyperparameters, e.g. the standard deviation parameter in SmoothGrad. Therefore, an essential question is how and based on what factors a practitioner should select an interpretation map and its hyperparameters from these established methods.

We argue that the “sample complexity” of estimating an interpretation map should be a factor in choosing the interpretation method and its hyperparameters. To understand the importance of sample complexity, we consider the following scenario from our CIFAR10 experiments in Section 6 where the expected norm difference between normalized SimpleGrad maps generated for two neural nets $f_{S_1},f_{S_2}$ learned under two *independent* training sets is $0.242$, while the same expected norm difference between normalized SmoothGrad maps (with $\sigma = 1$) is $0.019$. Then, a natural question is whether a practitioner should interpret and act based on the SimpleGrad map that would have been significantly different if the neural net had been trained using an independent training set, while the SmoothGrad map would behave with higher stability to the randomness of training data.

In this work, our goal is to bound the generalization error of the interpretation maps as a variance-based quantity with respect to the randomness in the training data. Specifically, our results show that the Simple-grad and Integrated-gradients maps could be highly affected by the randomness of training data, indicating that they may suffer from a high variance under a limited-size training set. On the other hand, we show SmoothGrad maps with sufficiently large variance will have a bounded generalization error, which suggests SmoothGard is a more stable and generalizable map with respect to the randomness of the training process.

To summarize our response, the importance of the generalization study for interpretation maps reduces to one’s answer to the following question: Should the variance of interpretation maps (with respect to the randomness of training data) be a factor for selecting an interpretation scheme and its hyperparameters from the long list of established methods in the literature? We believe the answer to this question is yes and our work aims to better understand the generalizability of gradient-based maps.

**2-The role of the population solution**  $f^*$ **in our generalization analysis**

**Re:** Please note that the saliency map $I(f^*,x)$ under the population solution $f^*$ is only a reference interpretation map that we use to quantify the variance of the interpretation map and compare it across training and test data. We note that if the minimizer to the population risk is not unique, we could define $f^*(x) = E_{f\sim \mathrm{Unif}(F^*)}[f(x)]$ as the expectation under the uniform distribution on the set of optimal population solutions $F^*$.

Furthermore, it can be seen that our generalization analysis will similarly hold if one substitutes the reference interpretation map $I(f^*,x)$ with any other reference map as long as that reference map is deterministic concerning the randomness of the training set of the neural net. For example, another practically relevant choice of the reference map could be $E_{S,A}[I(f_{A(S)} , x)]$ as the expectation of the reference map over a random training set $S$ of size $n$ sampled from the data distribution and the stochasticity of a training algorithm $A$ used to find the neural net weights $A(S)$. Then, our interpretation loss for a neural net $f_w$ with weights $w$ in the generalization analysis will be

$\mathcal{L}(f_w,x) = \Vert I(f_w,x) - E_{S,A}[I(f_{A(S)} , x)] \Vert $

In the above, we do not define the interpretation loss with respect to an “optimal neural net $f^*$” and instead measure the closeness of interpretation loss to the expected interpretation map over a random training set of size $n$. Please note that the same generalization results in Theorems 1,2 will hold under the above interpretation loss with a different reference point. We have added Remark 1 in the revised paper including this discussion.

---

### Meta-Review · Area_Chair_LBJZ · 2023-12-05

**Metareview:**

Currently, there are several concerns regarding the significance of studying the generalization of interpretation methods and the extent of the paper's theoretical contribution. The authors should make a revision and improve the motivation for their study.

**Justification For Why Not Higher Score:**

The main limitation of the work is a lack of proper motivation for studying the generalization of interpretation methods. The paper can potentially improve with further revisions.

**Justification For Why Not Lower Score:**

N/A

---

### Decision · Program_Chairs · 2024-01-16

Reject